Structural and evolutionary characteristics of dynamin-related GTPase OPA1

Li Dandan 1 2
Wang Jinlan 3
Jin Zichen 4
Zhang Zheng zhzhang.sdu@gmail.com zhangzheng@sdu.edu.cn 5
1 College of Biological Sciences, China Agricultural University , Beijing , China
2 National Institute of Biological Sciences , Beijing , China
3 Physical Examination Office of Shandong Province, Health Commission of Shandong Province , Jinan , China
4 Department of Chemistry, University of Minnesota , Minnesota, MN , USA
5 State Key Laboratory of Microbial Technology, Institute of Microbial Technology, Shandong University , Qingdao , China
Valencia Alfonso
Electronic publication date: 2019 Jul 8
Publication date: 2019
Volume: 7
Electronic Location ID: e7285
Received 2019 Jan 4; Accepted 2019 Jun 12
Copyright: ©2019 Li et al.
Copyright year: 2019
Copyright holder: Li et al.
License: This is an open access article distributed under the terms of the Creative Commons Attribution License, which permits unrestricted use, distribution, reproduction and adaptation in any medium and for any purpose provided that it is properly attributed. For attribution, the original author(s), title, publication source (PeerJ) and either DOI or URL of the article must be cited.
License URL: https://creativecommons.org/licenses/by/4.0/

Keywords: Protein–protein interaction, Modeled structure, OPA1, Disease-related sites, Bioinformatic analysis

Funding: China Postdoctoral Science Foundation 2018M642649 This work was supported by the China Postdoctoral Science Foundation (2018M642649). The funders had no role in study design, data collection and analysis, decision to publish, or preparation of the manuscript.

==============================
OPA1 is a dynamin-related GTPase that controls mitochondrial fusion, cristae remodeling, energetics and mtDNA maintenance. However, the molecular architecture of OPA1 is poorly understood. Here we modeled the structure of human OPA1 by the threading approach. We found that the C-terminal region of the OPA1 protein had multiple functional domains, while the N-terminal region was rich in alpha helices and did not include specific domains. For the short soluble forms of OPA1, we observed that there were obvious hydrophobic regions near the two cleavage sites and the N-terminal was positively charged after cleavage. The blue native analysis revealed that the protein could form stable homodimers. In addition, the evolutionary conservation of the C-terminal region, where most of the known mutated disease-related sites were located, was significantly higher than that of the N-terminal region. These findings provided new insights into the structure and biochemical function of OPA1.

Introduction

Mitochondria undergo a constant dynamic balance between organelles fusion and fission to maintain their network morphology and functions (Van der Bliek, Shen & Kawajiri, 2013; Wai & Langer, 2016). The functional relevance of mitochondrial dynamics has been emphasized by its requirement during embryonic development and its repercussions in the main functions of the organelle, including respiration, response to cellular stress, calcium homeostasis and apoptosis (Chan, 2012; Eisner, Picard & Hajnoczky, 2018; Noguchi & Kasahara, 2018). Several conserved GTPase mediate the mitochondrial dynamics: mitofusins (mitofusin 1 and mitofusin 2) control fusion of the outer mitochondrial membrane (OMM) while Drp1 is involved in fission of OMM (Labbe, Murley & Nunnari, 2014; Pernas & Scorrano, 2016); the conserved dynamin-related GTPase optic atrophy 1 (OPA1) is indispensable for both cristae morphology and the inner mitochondrial membrane (IMM) fusion (Belenguer & Pellegrini, 2013; Del Dotto et al., 2018a; Pernas & Scorrano, 2016).

Many evidences indicate that OPA1 is associated with other important mitochondrial functions, including mitochondrial DNA (mtDNA) maintenance, which is probably anchored to the IMM, and the oxidative phosphorylation efficiency (Belenguer & Pellegrini, 2013; Del Dotto et al., 2018a). In mouse embryonic fibroblasts, global loss of OPA1 results in fragmentation of the mitochondrial network, together with critically reduced mtDNA copy number, as well as significant disorganization of cristae structure and a reduction of respiratory capacity (Chen et al., 2010; Cogliati et al., 2013; Song et al., 2007; Song et al., 2009). Re-expression of any OPA1 isoform can restore energetic efficiency, mtDNA content, and cristae structure (Del Dotto et al., 2017). OPA1 oligomerization, tightening cristae junctions, is required for apoptosis regulation by maintaining the cytochrome c inside the cristae and controlling its release (Frezza et al., 2006; Olichon et al., 2003; Yamaguchi et al., 2008). In addition, OPA1-dependent modulation of cristae structure is necessary for cellular adaptation to energy substrate availability (Patten et al., 2014).

Mutations in OPA1 cause the disease dominant optic atrophy (DOA), one inherited optic neuropathy that is characterized by selective degeneration of retinal ganglion cells and classically presents in early childhood with progressive visual failure (Del Dotto et al., 2018b; Lenaers et al., 2012). Biochemical studies indicated that OPA1 disease alleles associated with DOA displayed a variety of mitochondrial defects in several activities, involving GTP hydrolysis, cardiolipin association, and membrane tubulation (Ban et al., 2010; Belenguer & Pellegrini, 2013; Zanna et al., 2008). Compared with the classical optic atrophy, DOA and deafness (DOAD) associating neurosensory deafness and DOA plus involving in other clinical manifestations like myopathy, progressive external ophthalmoplegia or spastic paraplegia, are syndromic dominant optic atrophy (Amati-Bonneau et al., 2008; Del Dotto et al., 2018b; Lenaers et al., 2012; Yu-Wai-Man et al., 2010). Moreover, OPA1 mutations have been related to an expanding spectrum of neurodegenerative phenotypes, such as Behr-like syndrome, syndromic parkinsonism, dementia and others (Carelli et al., 2015; Del Dotto et al., 2018b; Marelli et al., 2011).

OPA1 belongs to the dynamin superfamily proteins. Dynamins are large GTPases, including classical dynamins (dynamin 1, dynamin 2, dynamin 3), mitofusins, Drp1, OPA1, Mx proteins, guanylate-binding proteins (GBPs) and atlastins in eukaryotic cells (Jimah & Hinshaw, 2019). The mitofusins, Drp1, OPA1 and atlastins are involved in the process of membrane remodeling; the classical dynamins mainly function in the clathrin-mediated endocytosis and budding of vesicles; additional dynamins like GBPs and Mx proteins restrict pathogens (Anderson et al., 1999; Haller et al., 1981; Pernas & Scorrano, 2016). Several structures of dynamins have been solved so far (Jimah & Hinshaw, 2019). The structures show that all dynamins include a GTPase domain that binds and hydrolyzes GTP and an α-helical bundle domain. Most dynamins also contain a middle domain involved in oligomerization and a GTPase effector domain (GED) that are associated with stimulation of GTPase activity. Several dynamins contain a domain that can be a transmembrane domain, a sequence or a pleckstrin-homology domain (PH domain) for interacting with lipid membranes (Praefcke & McMahon, 2004). However, due to its exceptional complexity, there is still no crystal structure of the OPA1 protein.

In human, the OPA1 gene is built from 30 exons and has been reported to generate at least eight mRNA variants by the alternative splicing of exons 4, 4b and 5b (Fig. 1A). OPA1 is a dynamin-related GTPase with a mitochondrial targeting sequence (MTS), followed by a transmembrane (TM), which need to be further cleaved to execute mitochondrial function (Belenguer & Pellegrini, 2013). Precursors translated from the eight OPA1 mRNA are targeted to mitochondria via MTS, which is cleaved by the mitochondrial processing peptidase (MPP) to give rise to the long forms (l-forms) anchored to IMM. About half of l-forms are then further proteolytically processed by OMA1 and YME1L proteases to produce the short forms (s-forms), which are soluble in the mitochondrial intermembrane space (IMS) (Anand et al., 2014; Ishihara et al., 2006; MacVicar & Langer, 2016; Song et al., 2007). Remarkably, the four isoforms including exon 4b are completely processed into s-forms (Song et al., 2007). Although OPA1 protein have been studied for many years, the molecular architecture of OPA1 must be elucidated to better understand its functions. Here, we modeled the structure of the human OPA1 by the threading approach, analyzed the Protein–protein interaction, and explored the characteristics of the sequences and structures through bioinformatics analysis.

Figure 1 Schematic representation of human OPA1 isoforms.

(A) In human, OPA1 gene is built from 30 exons, three of which (4, 4b and 5b) are alternatively spliced leading to eight variants (isoforms 1–8). OPA1 exons (numbers) were schematized by the short line. Mitochondrial proteolytic cleavage sites for mitochondrial processing peptidase (MPP), OMA1 (S1) and YME1L (S2) were indicated. (B) Average hydropathicity of OPA1 N-terminal exon-encoded peptides. (C) The pI value of OPA1 N-terminal exon-encoded peptides. Proteolytic cleavage at S1 and S2 sites by the OMA1 and YME1L peptidase generates the IMS soluble s-forms. Exon 5-encoded peptide was divided into two parts by site S1, while the N-terminal part was labeled with 5N and the C-terminal was labeled with 5C. The same labeling method was used for exon 5b-encoded peptide cleaved at site S2.

Materials & Methods

OPA1 information acquisition

The information of human OPA1 gene, exons and isoforms were obtained from the NCBI and Ensemble databases (Zerbino et al., 2018). The other vertebrate OPA1 protein sequences were acquired from the NCBI reference sequence (RefSeq) database (O’Leary et al., 2016). All partial sequences and low quality proteins were excluded. The crystal structures of proteins from dynamin family were obtained from the PDB database (Rose et al., 2017).

Sequence analysis

The theoretical isoelectric points and the grand average hydropathicity were calculated by ExPASy server (Bjellqvist et al., 1993). The eight human OPA1 isoforms and all vertebrates OPA1 proteins were aligned with MAFFT (Katoh & Standley, 2013), respectively. Using the longest splice variant OPA1 in vertebrates, the evolutionary conservations of amino acid residue positions in the OPA1 sequences was measured by using ConSurf algorithm (Ashkenazy et al., 2016). The best evolutionary substitution model was used and calculation was based on the Empirical Bayesian paradigm. The sequence and modeled structure of human OPA1 were used to show the nine-color conservation grades. Sequence logos were generated as graphical representations of the multiple sequence alignments of the amino acids (Crooks et al., 2004). Phylogenetic analyses were conducted using MEGA version 7 by the bootstrap neighbor joining method (Kumar, Stecher & Tamura, 2016).

Structural analysis

I-TASSER was used to predict the structures of the human OPA1 protein (Yang et al., 2015). The structural model was refined by fragment-guided molecular dynamics (FG-MD) simulations at the atomic-level (Zhang, Liang & Zhang, 2011). The quality assessment of Ramachandran plot had been used to quantitatively assess the accuracy of protein structure predictions. The statistical data of the Ramachandran plot was calculated by PROCHECK (Laskowski et al., 1993). The protein structures were displayed by PyMol (Schrödinger, LLC). Pairwise structural differences between human OPA1 and other proteins from dynamin family were measured by TM-align (Zhang & Skolnick, 2005). The structural similarity of two protein structures was measured with the TM-score that had the value (0, 1) (Xu & Zhang, 2010). The higher the value was, the more similar the two aligned structures were. The two modeled structures of human OPA1 protein were taken as target 1 and target 2, respectively, to submit to PRISM protein–protein docking server for predicting possible interactions and how the interaction partners structurally connect (Baspinar et al., 2014).

Protein purification and blue native page

A human OPA1 construct (exons 6–28) was expressed in Escherichia coli BL21 (DE3) cells as N-terminal His6-SUMO-tagged fusion proteins using the pET28a plasmid (His6-SUMO-OPA1). Cells were grown in LB medium at 37 °C to an OD600 of ∼0.6. Protein expression was induced by addition of 100 µM IPTG and cultures were incubated overnight at 16 °C. The cell pellet was resuspended in 10 mM Tris-HCl pH 8.0, 20 mM imidazole and 500 mM NaCl buffer. Cells were lysed using a microfluidizer (Microfluidics). After centrifugation at 12,000 rpm for 1 h at 4 °C, the soluble extract was filtered and combined with Ni-NTA agarose (GE Healthcare) in batch. After beads were packed into a column, the combination was washed with resuspension buffer supplemented with 40 mM imidazole, and protein was eluted with an imidazole gradient to a final concentration of 500 mM. Peak fractions were pooled, and the His6-SUMO-tag was cleaved by ULP1 peptidase in the 4 °C overnight to generate the OPA1 (exons 6–28) protein. Then diluted the NaCl concentration to 50 mM and applied to Resource Q column (GE Healthcare). Peak fractions were concentrated and loaded to a Superdex 200 10/300 gel filtration column (GE Healthcare) using the buffer with 10 mM Tris-HCl, pH 8.0 and 150 mM NaCl. The protein was concentrated to ∼10 mg/ml prior to freezing. All the samples were analyzed by SDS-PAGE electrophoresis. The procedure for SDS-PAGE was based on a general protocol as previously described (Simpson, 2006). All the samples heated at 95 °C for 10 min and loaded onto a polyacrylamide gel that was consist of 5% stacking and 10% resolving gel, which was run at 80 V for 0.5 h, and then at 100 V for 1 h.

Blue native page was performed as described previously with slight modifications (Wittig, Braun & Schagger, 2006). 10 mg of the OPA1 (exons 6–28) protein was loaded into a blue native page gel that was consist of 3.5% stacking and 4–10% gradient separating gel. The cathode buffer (7.5 mM imidazole with pH 7.0, 50 mM tricine, supplemented with 0.02% w/v coomassie brilliant blue G250) and the anode (25 mM imidazole with pH 7.0) were chilled at 4 °C before used. Then, electrophoresis was started at 80 V for 30 min and adjusted to 120 V for 10 h.

Results

Sequence analysis of human OPA1 isoforms

The human OPA1 gene contains eight spliced variants (Fig. 1A). Proteolytically processed by OMA1 and YME1L proteases in the domains were corresponding to exons 5 and 5b, containing the cleavage sites S1 and S2, respectively. In principle, each mRNA variant form can produce a long isoform and one or more short isoforms. However, the four isoforms (3, 5, 6 ,8) including exon 4b completely processed into s-forms. We confirmed that exon 4b-encoded peptide was hydrophobic while the others around were hydrophilic (Fig. 1B), which was consistent with the previous study (Elachouri et al., 2011). The hydrophobicity of exon 4b-encoded peptide might be helpful to recruit the OMA1 protease, thus, promoting the cleavage at site S1. Although exon 5b-encoded peptide was hydrophilic as a whole, there was a hydrophobic region just before site S2, hypothesizing that the hydrophobic segment can associate with YME1L protease to promote the cleavage at site S2.

We also noticed that the N-terminus of OPA1 s-forms were positively charged (Fig. 1C). The isoelectric point (pI) value of exon 5-encoded peptide was 6.8, while the pI value of the remaining peptide segment increased to 8.6 after being cleaved at site S1. Similarly, the pI value of exon 5b-encoded peptide rose from 6.8 to 9.5 after the site S2 was cut. This result revealed that the N-terminal of OPA1 s-forms all had positive charge no matter it was cleaved at site S1 or S2.

Structure characteristic of human OPA1

Taking the longest splice variant 8 including 30 exons as an example, we used the threading approach to model the three-dimensional structure of human OPA1 protein excluding MTS and TM regions (Fig. 2). According to quality assessment, the accuracy of the OPA1 protein structural models were acceptable (Fig. S1). The structure showed that OPA1 protein could be clearly divided into two regions, the N-terminal and C-terminal region. The N-terminal region (encoded by exons 3–8) was rich in alpha helix, which did not include specific domains. The difference among the 8 spliced variants of OPA1 protein, l-forms and s-forms was the region of N-terminal. The N-terminal region of l-forms contained 138–228 amino acid residues, while s-forms included only 104–134 residues.

Figure 2 The three-dimensional structure of human OPA1 protein.

(A) Structure-based domain architecture of OPA1 isoform 8. The domain assignment was indicated below. Each domain was labeled by different colors, corresponding to the schematic representation of exons. (B) Modeled structure of isoform 8 excluding MTS and TM (long form). The N-terminal region was rich in alpha helix and did not include specific domains, while the C-terminal region of OPA1 protein was a dense structure containing multiple domains. (C) Modeled structure of human OPA1 protein (short form).

The C-terminal region of OPA1 protein was a dense structure containing multiple domains and was identical for all isoforms. In the human OPA1 protein, each domain of the C-terminal region had perfectly corresponded with exons. The GTPase domain corresponded with exons 9–16, middle domain corresponded with exons 19–22 and a GED corresponded with exons 26–28. The peptide encoded by exons 17–18 formed a long helix that connected GTPase domain and middle domain. The exons 23–25 encoded a peptide, a PH domain, located between middle domain and GED.

The PH domain of classical dynamins is responsible for their interaction with negatively charged lipid membranes (Praefcke & McMahon, 2004). The pI analysis indicated that the pI value of PH domain of human OPA1 protein was 7.7, especially the pI value of exon 25-encoded peptide was as high as 9.5 whereas the pI value of the middle domain and GED next to PH domain were 5.0 and 6.8, respectively. Thus, the PH domain of OPA1 carried positive charge, might be involved in interaction with negatively charged phospholipid molecules.

Comparing the structure of the C-terminal region of human OPA1 protein with the nine-known structures of the dynamin superfamily proteins (Fig. 3), the results showed that OPA1 and other human dynamins had two common domains: a GTPase domain and an α-helical bundle. Interestingly, human OPA1 protein was similar to dynamin 1, dynamin 3, Drp1, MxA and MxB proteins, and the TM-score of all was higher than 0.5, although these proteins did not mediate membrane fusion like OPA1. Comparing OPA1 with mitofusin 1, atlastin 1 and altastin 3, which were involved in membrane fusion, the TM-score of all was minor than 0.3. In addition, similar to dynamin 1, dynamin 3 and Drp1, OPA1 had a PH domain, while other proteins involved in fusion process did not have. Significant structural differences indicated that OPA1 might have a unique mechanism for controlling inner membrane fusion.

Figure 3 Comparison between the structure of OPA1 protein and other human dynamins.

(A) Phylogenetic tree of the human dynamin superfamily members. The structure of OPA1 protein was compared with that of dynamin 1 (B), dynamin 3 (C), Drp1 (D), MxA (E), MxB (F), mitofusin 1 (G), GBP1 (H), atlastin 1 (I) and atlastin 3 (J) separately. Phylogenetic analyses were conducted by the bootstrap neighbor joining method. The tree was drawn to scale, with branch lengths in the same units as those of the evolutionary distances used to infer the phylogenetic tree. The structure of OPA1 was showed in a modeled structure, while dynamin 1, dynamin 3, Drp1, MxA, MxB, mitofusin 1, GBP1, atlastin 1 and atlastin 3 were displayed by crystal structures (PDB code: 3snh, 5a3f, 4bej, 3szr, 4whj, 5yew, 1dg3, 3q5e and 5vgr). The TM-score calculated according to the comparison results was also displayed.

Dimerization of human OPA1 protein

To determine the oligomeric form of OPA1, we expressed the common part of the eight spliced variants of human OPA1 (exons 6–28) protein in E. coli. The protein was detected by SDS-PAGE electrophoresis (Fig. S2). From the peak position of the OPA1 (exons 6–28) elution profiles of a S200 size-exclusion column, the result indicated that the molecular weight of the protein was about 170 kD, thus, we speculated the protein may exist as dimers in solution. Additionally, in blue native electrophoresis, we found nucleotide-free OPA1 formed stable dimers and multiple oligomers, indicating that the OPA1 molecule itself could be interacted and formed a stable multimer.

Further, we predicted the possible mode of dimerization of the C-terminal region by using the human OPA1 protein modeled structure. Protein–protein docking results showed that the C-terminal region of the human OPA1 protein could form two stable homodimers modes. In the first mode of homodimer, the GTPase domains of the two monomers were approximately 130 Å apart, and the dimeric interface involved in 23 pairs of interaction among 15 amino acid sites, 10 pairs of which were from the interaction between the linker domain and middle domain, 12 pairs of which were from the interaction between the middle domain and the GED, while one pair of sites was between the two same GED domains (Fig. 4A). In the second homodimer, the GTPase domains of the two monomers were approximately 100 Å apart, and the dimeric interaction interface involved in 33 pairs of interaction among 18 amino acid sites, 17 pairs of which were from the interaction between two middle domains, 12 pairs of which were from the interaction between the middle domain and the GED, and four pairs from the interaction between the two GED domains (Fig. 4B). These results indicated that, similar to dynamin 1 and MxA, OPA1 could assemble to dimers and higher-order oligomers via middle domain and GED, which was different from mitofusins that mediated membrane fusion by GTPase domains (Cao et al., 2017). Additionally, interactions between GTPase-GTPase domains had been reported to be essential for the function of dynamins, but our results showed neither of these two dimeric interfaces of nucleotide-free OPA1 proteins involved in the GTPase domain.

Figure 4 Prediction of OPA1 homodimeric interaction.

(A) One stable mode of dimerization. The homodimerization interface had been rotated. (B) The other possible mode of dimerization. The model of OPA1 homodimeric interaction was predicted through docking calculations. The right was inter-residues interaction in potential dimerization interfaces. The interaction partners were connected by broken lines. The modeled structure of the OPA1 C-terminal region was used for docking analysis. The residues were numbered according to the human OPA1 isoform 8.

Conserved sites in OPA1 protein

Furthermore, we collected all the vertebrate OPA1 proteins by sequence similarity search. Totally, more than 900 of OPA1 protein sequences were discovered from 205 species including several spliced variants. The sequence alignments was shown in the form of sequence logo map (Fig. S3). Using the longest spliced variant of OPA1 protein sequence in each species of vertebrates, we analyzed conserved amino acid sites in the OPA1 proteins and displayed them through the modeled structure of the human OPA1 protein (Fig. 5A).

Figure 5 Evolutionary conservation of vertebrates OPA1 protein sequence.

(A) Mapping of evolutionary conservation of amino acid positions in the OPA1. The conservation scale was defined from the most variable amino acid positions (grade 1, colored turquoise) which were considered as rapidly evolving, to the most conservative positions (grade 9, colored maroon) which were considered as slowly evolving. The nine-color conservation grades were mapped onto the structure of human OPA1. (B) Distribution of evolutionary conservation in human OPA1 gene according to the exons. The highest percent of maroon represented the most conserved position in a protein.

The results showed that the N-terminal region of OPA1 protein was significantly less conservative than the C-terminal region (Fig. 5B). For those sites with the highest conservation (grade 9), the N-terminal region only occupied 27% of all sites and 66% in the C-terminal region. For those sites with the lowest conservation (grade 1), 43% of all sites were in the N-terminal region and 10% in the C-terminal region. Specifically, in the N-terminal region, the exon 4 and exon 5-encoded peptides had the lowest conservation while the peptides encoded by exons 6–8 had a higher conservation. In the C-terminal region, the conservation of the GTPase domain and the linker encoded by exons 17–18 were the highest, while the PH domain was less conservative. These results indicated that the linkers encoded by exons 17–18 and exons 6–8, next to the GTPase domain, may be functionally important. A large swing in the linker during the GTPase cycle might cause the power-stroke that led to fusion. However, being the PH domain less conserved, it was speculated that its function might vary from species to species.

Disease-related sites in human OPA1 protein

Human OPA1 mutations have been associated to a large spectrum of neurodegenerations. 171 amino acid site variants in the human OPA1 had been analyzed by the locus-specific database dedicated to OPA1 (Data S1). The positions of the amino acid sites in human OPA1 protein were in accord with those of spliced variant 8. These variations include substitutions, duplications, deletions and insertions, but do not include synonymous mutations, nonsense mutations, and frameshift mutations. Of the 171 variants, 136 were associated with diseases, 10 were not related to diseases and 25 were variants of unknown significance. About 80% of disease-related variant sites were the highest conservative in the vertebrate OPA1 proteins (grade 9), while only 30% of the variant sites unrelated to disease had the corresponding values. Besides, 60% of variant sites that were not related to diseases had the lowest conservation (grade 1), while less than 10% of disease-related variant sites were the lowest conservative. Therefore, disease-related mutations in the OPA1 protein occurred mainly at sites with highly evolutionary conservation.

The N-terminal region of OPA1 protein occupied 34% of the total length, but the disease-related mutation sites located in this region accounted for only 15% of the total disease-related mutation sites in the OPA1 while the mutations unrelated to diseases in this region occupied 70% of the total mutation sites unrelated to diseases. The disease-related variant sites in the N-terminal region were concentrated in the exons 1–2 coding region (8 variants) and the exons 6-8 coding region (11 variants), while the variant sites unrelated to disease were concentrated on the exons 4, 4b and 5b (five variants). For the C-terminal region, more than 25% of the sites in the GTPase domain and the linker region encoded by exons 17–18 were found to have disease-related mutations, respectively. And about 20% of sites in the GED were also found to present disease-related variation. Therefore, most of the disease-related mutations in the OPA1 protein were located in the C-terminal region, while in the alternative splicing region, almost no disease-related mutation occurred.

Discussion

OPA1 is a member of dynamin superfamily, which is essential for shaping the cristae morphology and IMM fusion (Belenguer & Pellegrini, 2013; Praefcke & McMahon, 2004). In human, the OPA1 proteins include soluble s-forms and l-forms containing the TM region. However, there is a lack of understanding of the whole structure of OPA1 and the structural differences among various spliced variants, l-forms and s-forms. In our study, the modeled structure of human OPA1 protein revealed that its structure was divided into N-terminal and C-terminal regions. N-terminal region did not contain the specific domains and was structurally a long peptide chain rich in alpha helices. The length of the long peptide chain in the N-terminal region was the only difference among the eight OPA1 isoforms, while there was no difference in the functional domains. For the OPA1 s-forms, it lacked the TM region and the length of the peptide chain in the N-terminal region was shorten. More interestingly, we found that the N-terminal of exons 5 and 5b encoding peptides were positively charged after cleavage, indicating that OPA1 s-forms may interact with the negatively charged phospholipid molecules in the membrane.

The C-terminal region of OPA1 protein was a dense structure comprising a GTPase domain, a middle domain, a PH domain and a GED. The structure of this region was different from mitofusins and atlastins that mediated membrane fusion, while it was similar to dynamin 1 and MxA. The OPA1 protein contained a PH domain that interacted with phospholipid molecules, while mitofusins and atlastins did not include. In addition, similar to dynamin 1 and MxA, OPA1 could self-assemble to form dimers through the middle domain and GED, while this was also different from mitofusins and atlastins that dimerized by GTPase-GTPase domains. All these findings indicated that OPA1 might have a specific fusion mechanism.

The prior studies indicated that a homotypic OPA1 interaction, tightening cristae junctions, mediated membrane tethering (Ban et al., 2017; Frezza et al., 2006; Olichon et al., 2003). Our modeled structural analysis indicated that the C-terminal region of human OPA1 protein could form stable homodimer. Although the obvious difference between the eight OPA1 isoforms was concentrated on the N-terminal region, all of them maybe adopt the same mode by trans-OPA1 interaction when maintaining the cristae morphology. The OPA1 s-forms could also form a trans-interaction, and its N-terminal might interact with cardiolipin in the membrane. An in vitro membrane fusion assay unveiled that OPA1 l-form on one side of the membrane and cardiolipin on the other side, are the minimal components sufficient and necessary for fusion (Ban et al., 2017; Liu & Chan, 2017). According to the electric charge analysis, the PH domain had a strong positive charge region while cardiolipin had a negative charge region. Furthermore, a deletion mutant assay suggested that the domain next to the GTPase domain was necessary for cardiolipin binding (Ban et al., 2017; Del Dotto et al., 2018a). Combined with our work, we speculated that the charge interaction between PH domain and cardiolipin promoted binding and then accelerated double-membrane fusion. In short, these findings could be useful to better understand the biochemical functions of OPA1.

Conclusions

In this work, we modeled the whole structure of human OPA1 protein, revealing that its structure was divided into N-terminal and C-terminal region. The N-terminal region was rich in alpha helices and did not include specific domains. The eight OPA1 spliced variants only differed in the length of the long peptide chain in the N-terminal region but not in the functional domains. By contrast, the C-terminal region of OPA1 protein was a dense structure containing a GTPase domain, a middle domain, a PH domain, and a GED. The structure of the C-terminal region of OPA1 protein was not similar to that of other dynamin superfamily members that mediated membrane fusion, while it was similar to dynamin 1 and MxA, which could self-assembled to form dimers by the middle domain and GED. Additionally, the evolutionary conservation of the C-terminal region was significantly higher than that of the N-terminal region and the known mutated disease-related sites were mostly located in the C-terminal region of OPA1 protein. Overall, these findings provide novel insights into the structural and evolutionary characterizations of OPA1.

Supplemental Information

Figure S1 The quality assessment of Ramachandran plot had been used to quantitatively assess the accuracy of human OPA1 protein structure predictions

The statistical data of Ramachandran plot was calculated by PROCHECK. The results showed that the proportion of core% + allow% (the core residues: residues in most favoured regions; allow residues: residues in additional allowed regions) was more than 90% for the protein structure of all predicted OPA1 domains. While, the ratio of all disallowed residues (residues in disallowed regions) was less than 5%. According to quality assessment, the accuracy of the OPA1 protein structural models were acceptable.

Click here for additional data file.

Figure S2 Preparation of recombinant human OPA1 protein using E. coli

(A) The OPA1 (exons 6-28) elution profiles of a S200 size-exclusion column. (B) SDS-PAGE and Blue native PAGE analysis of purified human OPA1 (exons 6-28) protein. Purification was carried out more than 10 times for OPA1 with similar purity and yield.

Click here for additional data file.

Figure S3 Multiple sequences alignments of vertebrates OPA1 protein sequences

Sequence logos were generated as graphical representations of the multiple sequence alignments of the amino acids. The positions of the amino acid sites in vertebrates OPA1 protein are in accord with those of human OPA1 isoform 8.

Click here for additional data file.

Data S1 The information of known disease-related sites of the human OPA1 protein

Click here for additional data file.

We acknowledge Professor Xiangshu Jin of Michigan State University for insight and advice on biochemical assays.

Additional Information and Declarations

Competing Interests

Author Contributions

Data Availability

The authors declare there are no competing interests.

Dandan Li and Zheng Zhang conceived and designed the experiments, performed the experiments, analyzed the data, prepared figures and/or tables, authored or reviewed drafts of the paper, approved the final draft.

Jinlan Wang and Zichen Jin analyzed the data, authored or reviewed drafts of the paper, approved the final draft.

The following information was supplied regarding data availability:

The raw data are available as Supplemental File.

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
