# Peer review of "Structural and evolutionary characteristics of dynamin-related GTPase OPA1"

_PeerJ, doi:10.7717/peerj.7285_

## Round 0.1 · original submission · Minor Revisions

Dear Authors

As you can see the comments are quite positive and point to a number of aspects that require clarification or better explanations, including a better adjustment between results and conclusions.

Particularly important are the missing details on modelling and alignments.

·

Basic reporting

The authors with this papers aimed at elucidating the molecular architecture of OPA1 to better understand the IMM fusion, by modeling the structure of human OPA1 and analyzing the protein-protein interaction and the sequence/structure characteristic through bioinformatics approach.

The English is professional and mostly clear, with some little errors.
Line 24: I'm not sure I understood correctly what the authors mean writing “threading approach”.
Line 51: reduction instead of reducation
Line 141: “there was a hydrophobic region was found just before site S2”. Too much “was” in the phrase.
Line 162: “The exons 23-25 encoded peptide”, an “a” is missing after encoded.
Line 202: “In additional”, probably “In addition” or “Additionally” will sound better.
Lines 220: “However, the PH domain was less conserved, it was speculated that its function might vary from species to species”, probably “However, being the PH domain less conserved, it was speculated that its function might vary from species to species” could sound better. Or something likes this.
Line 219: linkers instead of linker

Sufficient field background is provided with adequate literature. Only some suggestions:
Lines 41-44 “In mammalian cells, the conserved dynamin-related GTPase optic
atrophy 1 (OPA1) is indispensable for both cristae morphology and the inner mitochondrial
membrane (IMM) fusion (Friedman & Nunnari 2014; McNew et al. 2013; Pernas & Scorrano
2016)” The references Friedman & Nunnari 2014 and McNew et al. 2013 are really nice reviews but do not exactly treat the topic of the sentence. Friedman & Nunnari is very focused on fission rather than fusion. McNew is focused on GTP-dependent membrane fusion rather then OPA1.

Lines 59-60: “Biochemical studies indicated that OPA1 disease alleles associated with DOA displayed a variety of mitochondrial defects in several activities, involving GTP hydrolysis, cardiolipin association, and membrane tubulation (Ban et al. 2010; Belenguer & Pellegrini 2013)” I suggest to add a reference here regarding the different degree of mitochondrial dysfunctions observed in patients’ cells.

Line 63-64: regarding the DOA+ phenotypes, Ban 2010 is not the correct reference. For a short summary about all the possible phenotypes caused by OPA1 mutations I suggest Del Dotto et al. 2018 Pharm Res. For a more in-depth knowledge I suggest the experts in the field: Yu-Wai-Man, Chinnery, Carelli, Amati-Bonneau. For example Yu-Wai-Man 2010 Brain described that DOA+ phenotypes could be generated by mutations in dynamin domain or other domains, and not only in GTPase domanin, and that those mutations could be missense, but also non sense or deletion or splice. Furthermore, DOA+ phenotypes could be given also by mutations that generate haploinsufficiency. Thus the phrase “The DOA+ phenotypes are missense mutations, relating to the highly conserved amino acid residue positions and exerting a dominant negative effect on multi-systemic disease” should be revised.

The structure of the article conforms to an acceptable format of ‘standard sections’.

Figures are relevant to the content of the article and of sufficient resolution. They are described quite appropriately in text, but the legends should be detailed better. For example in figure legend 2 the authors should detail what N or C does mean.
The legend of the supplementary table 1 is missing. For a better understanding by readers, the authors should detail which OPA1 variant are using (variant 8 or 1) and in the first column specify if is aminoacid position or nucleotide position.

The raw data has been made available.

The results are not completely relevant to the hypotheses.
Already in the title and in the abstract the authors declare that their results reveal something more about the mechanism of mitochondrial fusion OPA1 dependent, but in my opinion it is not so. Some data are of interest but do not allow to make new assumptions about the fusion mechanism.
For example, concerning the fusion Mitofusins dependent, Qi, 2016 JCB and Mattie 2018 JCB, combining bioinformatic predictions from the Mitofusins structure with biochemical and functional experiments, really provide structural insights into the mechanisms of mitochondrial outer membrane fusion. The work by Qi et al. further implicates the GTPase domain as the site of mitochondrial tethering, rather than the C-terminal HR2 domain as previously proposed. The work of Mattie et al. shows that metazoan Mitofusins contain only a single transmembrane domain and leads to a revised understanding of Mfns as single-spanning outer membrane proteins with an N(out)–C(in) orientation. Furthermore Mattie et al. provide functional insight into the IMS contribution to redox-regulated fusion events.
The authors, by bioinformatics analysis of the OPA1 structure, here presented some new information, as the possible mode of OPA1 dimerization clearly indicating the involved aminoacids. On the other hand much of the information reported is already known.
In figure 8 the authors proposed a model for cristae formation and membrane fusion that it is not well supported by functional experiments already present in literature and not well described/discussed by authors themselves.

Experimental design

Research question will be relevant and meaningful. Discovering something new about the structure of OPA1 that could help the understanding of the IMM fusion mechanism would really fill an existing knowledge gap. Unfortunately the results of the authors do not cover up this gap.

The investigations performed are rigorous.

Methods are described with sufficient detail and information to replicate. Only some errors and some suggestions:
Lines 112-113 “Blue native page was performed as described previously with slight modifications
(Wittig et al. 2006)” the authors should describe the modifications.
Line 112 1 mM GTP instead of 1mm GTP
Fig 5A showed an SDS-PAGE and this technique is not reported in Methods.
Lines 178-186 this part could be moved to Methods. The authors should detail better the purification conditions, as the amount of IPTG used, the elution conditions, temperature necessary for the expression of the protein….

Validity of the findings

Results section should be slightly reorganized.

Lines 121-138 are not results. This information is already present in literature. This part should be moved to the introduction section.

Lines 135 References Anand et al. 2014; MacVicar & Langer 2016 concern only the processing of OPA1 and not all the other notions. The authors should add some other references.

Figure 1 should be part of the introduction.

Lines 144-147: Concerning the different OPA1 functions, what could be the biological meaning of the different pI?

The paragraph on the dimerization of OPA1 is really interesting.
Lines 187-189: in figure 5B the authors show that OPA1 formed stable dimers and multiple oligomers independently on GTP/GDP binding, suggesting that OPA1 itself could interact and form oligomers. This result is really surprising, also because contradicts the literature knowledge.
Yamaguci 2008 Mol Cell reported that the addition of 1 mM GTP enhanced self-binding of Opa1, whereas 1 mM GDP completely blocked complex formation. Thus GTP promoted in vitro assembly of Opa1, whereas GDP favored the dissociated state.
Ban 2017 Mol Cell reported that in native-PAGE, a major dimer band at a steady state was moved to form higher oligomers of approximately 10 molecules after incubation with GTP or GTPγS, suggesting that GTP binding stimulates oligomerization on membranes.
Mitofusin 1, a dynamin-like GTPase as OPA1, forms dimers in the presence of GTP but not of GDP.
Qi 2016 JCB reported that MFN1 forms a dimer when GTP or GDP/BeF3−, but not GDP or other analogs, is added.
Ishihara 2004 JCS reported that in cell lysates, the cytosolic domains of MFN self-associate only when GTP is added at an enzyme-active temperature.
Which is the authors’ explanation? Since these results contradict the present literature, I think the authors should support better these data and not liquidate them in 3 lines without an appropriate discussion. I think other experiments are necessary.

Lines 190-204: this analysis is really interesting.
I have a doubt: the aminoacids shown in Figure 6 do not coincide with those described in the text.
In mode 1 there are 15AA instead of the 12 described. In mode 2 there are 18AA instead of the 17 described. Could the authors explain me better?
It is really interesting that, unlike mitofusins that dimerize by GTPase domains, for OPA1 neither of these two dimeric interfaces involved the GTPase domain or the PH domain. These differences should be reported and this part should be expanded and discussed better.

Lines 202-205: “In additional, only a small number of sites in the two dimeric interfaces overlapped, suggesting that the human OPA1 protein could further form a multimer on the basis of dimerization”. What does it mean? It would be interesting to explain this sentence better, perhaps adding some hypotheses.

Lines 219-221: which could be the function of the linkers encoded by exons 17-18 and exons 6-8? Could the authors write some hypothesis? Some speculations? The same for the PH domain.

The last paragraph of the results session, Disease-related sites in human OPA1 protein, is very confusing and has many inaccuracies.
Line 223: why the authors reported only 65 mutations?
On the locus-specific database dedicated to OPA1 (http://opa1.mitodyn.org/) at least 414 OPA1 variants are listed, more than 60% are considered pathogenic and two-thirds of them are in the coding sequence, mainly located in the dynamin and GTPase domains.

Lines 224-233 and Supplementary table 1: I don’t understand the diseases nomenclature that the authors use. In literature, usually, there are non-syndromic DOA, DOA and neurosensory deafness (DOAD) and syndromic-DOA (DOAplus). Since 2010 OPA1 mutations have been linked to an expanding spectrum of neurodegenerative phenotypes, such as spastic paraplegia, multiple sclerosis-like syndrome, Behr-like syndrome and most recently syndromic parkinsonism and dementia. These are the monoallenic condition, where the patients have 1 allele WT and 1 allele mutated. Then there are the bi-allelic conditions, where the patients have 2 alleles mutated with 2 different mutations associated with complex and severe phenotypes.
In the table what is DOA+ and OPA1? What does it means? What is OPA1 and BEHRS? Which is the differences between (DOA+ and OPA1) and (OPA1 and DOA+)?
I think the authors should conform to the nomenclature commonly used in literature. To get a summary overview on the mutations OPA1 and related diseases I recommend reading the review Del Dotto 2018 Pharm Res and to deepen my advice is to read the papers of experts in the field as Yu-Wai-Man, Amati-Bonneau, Carelli, Chinnery….

Line 234: “Within the 65 of disease-related variants, we discovered…”, it is not a discovery, but an observation.
This paragraph does not present any result but only observations, most of which have been known for a long time. Moreover, these observations are based on 65 mutations, when today more than 400 are listed in the database indicated above. I do not understand what the end of this paragraph is. At least the analysis should include the whole mutation panel. This paper is not a review. The last paragraph should be revised.

The discussion section is too hasty and pulled away. There is no real discussion of what the authors observe in the models they generate and proposed in figure 8.

Figure 8A: the cristae formation by OPA1 l-form. Cristae are not formed solely with long forms. Del Dotto 2017 Cell Rep and Lee 2017 JBC showed that long and short have the same ability to organize cristae. Thus, the authors should revise this part of the proposed model.
Figure 8B bottom: membrane fusion model. “s-form should support a bridge between l-form and cardiolipin”, I don’t understand the meaning of this phrase.
If short forms facilitated the IMM fusion, then the fusion rate of the long + short together should be greater than that of the long ones alone. But it is not so. Long forms have fusogenic capacity but are not able to interconnect the mitochondrial reticulum which remains therefore fragmented to 100% in cells. While long and short together influence the morphology of the reticulum that becomes filamentous and interconnected. The authors should explain better the proposed model, keeping in mind also the functional experiments already present in the literature.
Lines 263-264: “interaction between PH domain and cardiolipin promoted binding and then facilitated double-membrane fusion”, even the short form have the PH domain and should have the same fusogenic capacity as the long ones, but they are very less efficient. What explanation do the authors give?

Lines 267: “these findings have deepened our understanding of OPA1 and likely will ultimately lead to improvements in human health”, I think this sentence is a little bit exaggerated. The results reported by the authors are not particularly innovative. There are quite interesting data that could be useful to better understand the IMM tethering by OPA1 and how the fusion process can take place. Although the authors need to broaden and better discuss various aspects. But these results certainly do not lead to improvements in human health.

Additional comments

The paper is not particularly innovative, but has some interesting and new ideas that need to be developed better.
My advice is to implement the paper following the above suggestions.

Reviewer 2 ·

Basic reporting

Introduction:

The introduction needs more details.
I suggest the authors to develop a little more the general context of the study. In their introduction the authors indeed soon focus on OPA1. Only a short sentence describes mitochondrial dynamics; the main proteins that control mitochondrial fission or outer membrane fusion are not quoted. Furthermore, it would also be interesting to add few informations regarding the function of mitochondrial dynamics.
The description of the structure (and the mechanisms) of the Dynamin Super Family, well reviewed in the recent publication of Jimah & Hinshaw Trends in Cell biology 2019, also need to be expanded.

I recommend to the authors to check carefully the bibliography. I suggest to add more recent reviews on mitochondrial dynamics than “Chan 2012” and “Van Der Blieck 2013” (lines 40-41). These reviews are very interesting but significant advances have been made with regard to this topic in the last past years (Del Dotto 2018 only concerns OPA1). The same remark can be made for “Lenaers 2012” for optic nerve atrophies (line 58). Furthermore, some references are not well quoted: “Friedman & Nunari 2014” (line 48) does not focus on OPA1, in “Song 2007 and 2009” (lines 48-51) the experiments were made in Mouse Embryonic Fibroblast and not “humans”, “Praefcke & MacMahon 2014” (line 55) concerns Dynamins and not OPA1 ….

There it appears to be some confusions and some missing informations:
-line 53: Cristae-tightening-related OPA1 function is important not only for apoptosis (add Yamaguchi Mol Cell 2008) but also for mitochondrial energetics (Patten Embo 2016).
-line 58: Does OPA1 mutations really causes blindness in most cases” ?
-line 63: mtDNA deletion also occurs in isolated DOA and not only in DOA+ (Yu-Wai-Man Brain 2009).
-line 64: DOA+ phenotype are proposed to exert a dominant negative effect dominant.

Methods :

I recommend to the authors to move the paragraph “lines 178-185” from results to material and methods and to describe more carefully their structural analysis (see below)

Results and Figures :

I suggest the authors to add more details on OPA1 proteolysis (in the introduction).

More than 300 mutations have been described in OPA1 (not only 65). I suggest the authors to enlarge their analysis and to add the reference “https://databases.lovd.nl/shared/genes/OPA1”.

Figures are relevant and well described.
Does OPA1 really contain a PH domain as shown in Fig. 1 ?
Is OPA1 really purified to homogeneity (Figure 2A is a result of Comassie blue staining and has to be moved in Supp data) ?
A Figure comparing the schematic representation of dynamin super family members (classical dynamin and Dynamin Related protein) would be very informative.


Discussion :

I recommend to the authors to integrate to their model recent data showing in other that s-OPA1 alone can sustain cristae structure (Del Dotto et al Cell Report 2017, Lee et al JBC 2017) and to develop the description of the works concerning the various functions/properties of OPA1 isoforms.

Experimental design

The model proposed for OPA1 sounds good for the core domains 9-16 to 26-28 due to the large number of corresponding proteins probably included in the template but no explanation was given to justify the accuracy of the model.
However, the model proposed for the elongated form seems suspicious at least in its N-terminal extremity because of the succession of short alpha helices covering the region corresponding to exons 3 to 5b giving rise to a long helical N terminal domain. A random sequence will always give rise to a small alpha helix by this approach. I suggest you to provide sequence alignments especially in support of Figure 7, which leaves anyway to suppose that the model cannot be reliable for exons 3 to 5b.

Validity of the findings

The conclusions given on the structure of the elongated form of OPA1 need deeper critical discussion and can’t be supported in absence of experimental evidences supporting this model. For instance, experiments on the elongated form could be realized by SAXS.

Additional comments

Using a threading approach the authors model the structure of the human OPA1 protein. This study could have be interesting because the structure of this protein has not been experimentally solved and since this information could help to better understand its functions. However major bibliographic and experimental concerns have to be solved before publication.

---

## Round 0.2 · Minor Revisions

Dear Authors,

As you can see the referee still has a substantial number comments mostly related with style, clarity and references. I am confident that this will represent straight forward revisions.

There are also a small number of comments that require more attention including the ones related to the interpretation of the type of interaction between PH domain and cardiolipin in relation to membrane fusion.

Therefore, we will go for a short minor revision version that should be closed in a few days.

·

Basic reporting

Title and Introduction
I appreciate the changes in the title, which now better reflects the contents of the paper, and in the introduction, which now is more complete.
Only some little errors:
Line 42 “while Drp1 involves in fission of OMM” should be changed in “while Drp1 is involved in fission of OMM”. Furthermore, Drp1 is involved in OMM and IMM fission and not only in OMM. It oligomerizes to generate a constriction ring around mitochondria.

Line 50-51 the references Song 2007 and Song 2009 regard mitochondrial network and cristae morphology but not mtDNA copy number or respiratory capacity of MEF OPA1 null.

Line 96 “the four isoforms including exon 4b completely processed into s-forms” should be “the four isoforms including exon 4b are completely processed into s-forms”

Experimental design

Methods
Lines 150-151 SDS-Page is only mentioned but not described. At least a reference should be indicated.

Validity of the findings

Results
Lines 165-166 “We discovered that exon 4b-encoded peptide was hydrophobic while the others around were hydrophilic”.
This was already demonstrated by Elachouri 2011 Genome Research. In that paper the authors showed that NT-OPA1-exon4b is tightly bound to the mitochondrial membrane fraction and it is embedded in the IMM (by a protease protection assay on permeated mitochondria). Elachouri et al. reported “NT-OPA1-exon4b structure is composed of two transmembrane (TM) domains, one common to all OPA1 isoforms, and one encoded by exon 4b, plus a 35-aminoacid intermediate domain without signature motif….In vertebrates, the second TM domain consisting in a 15-amino-acid-long hydrophobic sequence is encoded by exon 4b, whereas in yeasts and Drosophila, the second TM domain is systematically present in the unique isoform”.
Thus, the authors should change the sentence “We discovered that exon 4b-encoded peptide was hydrophobic while the others around were hydrophilic” with “We confirm that ….” adding also the reference Elachouri 2011 Genome Res.

Lines 201 “Compared the structure of the C-terminal region..” should be changed in “Comparing the structure…”

Line 205-206 “the TM-score was all more than 0.5, although these proteins did not mediate membrane fusion like OPA1” should be write more clearly. “the TM-score of all was higher than 0.5…”could be correct?

Line 206-207 “Compared OPA1 with mitofusin 1, atlastin 1 and altastin 3, which were involved in membrane fusion, the TM-score was all less than 0.3” should be changed in “Comparing…. the TM-score of all was minor than 0.3”

Line 208 “OPA1 had a PH domain, while other proteins did not have” which proteins? Probably the sentence should be “OPA1 had a PH domain, while other proteins involved in fusion process did not have”.
Line 215-216 “From the peak position of the OPA1 (exons 6-28) elution profiles of a S200 size-exclusion column, the result indicated the molecular weight…” should be changed in “From the peak position of the OPA1 (exons 6-28) elution profiles of a S200 size-exclusion column, the result indicated that the…”

Line 218-220 It’s a pity that the authors decided to remove the experiment regarding OPA1 oligomerization independently by the presence of GTP or GTP, instead of going into more detail on this point. It was a really new and interesting result.

Line 225 and 230 “Between 15 AA…” and “Between 18 AA…” should be changed with among.

Line 235 “which was different from mitofusins that mediate membrane fusion” this sentence is not complete. It could be better “which was different from mitofusins that mediate membrane fusion by GTPase domains”. Add also a reference about mitofusins dimerization.

Lines 235-239 “Additionally, interactions between GTPase-GTPase domains had been reported to be essential for the function of dynamins, but our results showed neither of these two dimeric interfaces of nucleotide-free OPA1 proteins involved in the GTPase domain. We speculated that GTP binding and self-assembly may promoted OPA1 GTPase-GTPase domains dimer formation to mediate inner membrane fusion”. I'm not sure I understood what the authors mean. This sentence should be better explained.

Lines 262 “In human OPA1, mutations have been found to cause diseases” should be changed in “Human OPA1 mutations have been associated to a large spectrum of neurodegenerations”

Line 278 “The disease-related variant sites in the N-terminal region are concentrated..” the authors used the pass form in all the manuscript.

Lines 261-285 despite the changes that the authors have made following my suggestions, I still think that this paragraph does not add anything new.
The authors performed this long analysis to conclude that “Therefore, most of the disease-related mutations in the OPA1 protein were located in the C-terminal region, while in the alternative splicing region, almost no disease-related mutation occurred”. Nothing of new. The same massage was already reported in Amati-Bonneau 2009 IJBCB and Ferrè 2009 Hum Mut.

Discussion
Line 321 “What’s more, a deletion mutant assay suggested the domain next to the GTPase domain is necessary for cardiolipin binding” should be changed in “Furthermore, a deletion mutant assay suggested that the domain next to the GTPase domain is necessary for cardiolipin binding”.

Line 322 The reference Del Dotto 2018a is not included in the references list.

Lines 317-319 “By using an in vitro membrane fusion assay, the experiments unveiled the molecular basis of most minimal IMM fusion that OPA1 l-form and cardiolipin on each side of the membrane separately are sufficient for fusion (Ban et al. 2017; Liu & Chan 2017)” Write better this sentence.

Lines 315-325 This part remains confused, unconvincing and pulled away.
In the first my revision I underline that even the short forms have the PH domain and should have the same fusogenic capacity as the long ones, but they are very less efficient.
In the rebuttal letter the authors answer was” The s-forms have the PH domain, so the s-forms may also interact with cardiolipin on the membrane. However, there is an obvious difference between the l-forms and s-forms, the s-forms do not include the transmembrane region. In addition, the inner membrane fusion is a complex process. The interaction may promote binding, but it does not mean that the s-forms should have the fusogenic capacity”.
My experience on OPA1 derives from 18 years of studies on OPA1 ko or mutated cells and I really know that the obvious difference between the l-forms and s-forms is that the s-forms do not include the transmembrane region. Furthermore I quantified the fusion rate in OPA1 mutant cells and I really know that inner membrane fusion is a complex process.
In the revised manuscript the authors write that "Combined with our work, we speculated that the charge interaction between PH domain and cardiolipin promoted binding and then accelerated double-membrane fusion” and nothing else.
I repeat the same observation: this speculation is not supported by in vivo experiments. s-forms have the PH domain and can bind cardiolipin but they have not fusogenic capacity (Del Dotto 2017 Cell Report). Furthermore, l-forms alone have the same fusion rate of the l- and s-forms together (Del Dotto 2017 Cell Report). Therefore the role of the PH domain in the fusion process does not seem convincing to me. It could have a more structural role.
The authors should ameliorate this part.

Figures
Figure 2 reported OPA1 isoform 8 in its long and short forms. But isoform 8 is completely processed in s-forms and the l-form is not present in vivo.

Additional comments

the manuscript requires other small revisions

---

## Round 0.3 · accepted · Accept

Thanks for completing the revision, including the small edits and the changes in the title and figures.